# Is Tissue Still the Issue? The Promise of Liquid Biopsy in Uveal Melanoma

**DOI:** 10.3390/biomedicines10020506

**Published:** 2022-02-21

**Authors:** Daniël P. de Bruyn, Aaron B. Beasley, Robert M. Verdijk, Natasha M. van Poppelen, Dion Paridaens, Ronald O. B. de Keizer, Nicole C. Naus, Elin S. Gray, Annelies de Klein, Erwin Brosens, Emine Kiliç

**Affiliations:** 1Department of Ophthalmology, Erasmus MC Rotterdam, 3000 CA Rotterdam, The Netherlands; d.p.debruyn@erasmusmc.nl (D.P.d.B.); n.vanpoppelen@erasmusmc.nl (N.M.v.P.); d.paridaens@oogziekenhuis.nl (D.P.); n.naus@erasmusmc.nl (N.C.N.); 2Department of Clinical Genetics, Erasmus MC Rotterdam, 3000 CA Rotterdam, The Netherlands; a.deklein@erasmusmc.nl (A.d.K.); e.brosens@erasmusmc.nl (E.B.); 3Erasmus MC Cancer Institute, 3000 CA Rotterdam, The Netherlands; 4Centre for Precision Health, School of Medical and Health Sciences, Edith Cowan University, Joondalup, WA 6027, Australia; a.beasley@ecu.edu.au (A.B.B.); e.gray@ecu.edu.au (E.S.G.); 5The Rotterdam Eye Hospital, 3011 BH Rotterdam, The Netherlands; r.verdijk@erasmusmc.nl (R.M.V.); r.dekeizer@oogziekenhuis.nl (R.O.B.d.K.); 6Department of Pathology, Section Ophthalmic Pathology, Erasmus MC Rotterdam, 3000 CA Rotterdam, The Netherlands; 7Department of Pathology, Leiden University Medical Center, 2333 ZA Leiden, The Netherlands

**Keywords:** PET/CT, CT, US, OCT, exosomes, non-invasive, survival

## Abstract

Uveal melanoma (UM) is the second most frequent type of melanoma. Therapeutic options for UM favor minimally invasive techniques such as irradiation for vision preservation. As a consequence, no tumor material is obtained. Without available tissue, molecular analyses for gene expression, mutation or copy number analysis cannot be performed. Thus, proper patient stratification is impossible and patients’ uncertainty about their prognosis rises. Minimally invasive techniques have been studied for prognostication in UM. Blood-based biomarker analysis has become more common in recent years; however, no clinically standardized protocol exists. This review summarizes insights in biomarker analysis, addressing new insights in circulating tumor cells, circulating tumor DNA, extracellular vesicles, proteomics, and metabolomics. Additionally, medical imaging can play a significant role in staging, surveillance, and prognostication of UM and is addressed in this review. We propose that combining multiple minimally invasive modalities using tumor biomarkers should be the way forward and warrant more attention in the coming years.

## 1. Introduction

Uveal melanoma (UM) is a rare but life-threatening disease. The incidence of UM is mildly increasing, which is primarily due to the increased detection of small tumors [1]. This coincides with a trend towards non-surgical, eye-sparing therapeutic options, in which no tumor tissue is available [2], as shown in Figure 1. About 50% of the patients will eventually develop metastases, which are usually fatal within one year [3,4].

Despite numerous initiated trials over the last decades, improvement in progression free survival (PFS) has been sparse. Some elongation of the time to metastases was reported after dendritic cell vaccination, in which high-risk (and monosomy 3) patients received vaccinations with dendritic cells transfected with glycoprotein 100 (gp100) and tyrosinase [5]. However, new systemic therapeutic options are still being explored. For example, immune checkpoint inhibitors and more specifically monoclonal antibodies that target programmed cell death protein 1 (PD-1) and cytotoxic T-lymphocyte-associated antigen 4 (CTLA-4) receptors and their ligands (e.g., Nivolumab (anti-PD-1 antibody) and Ipilumab (anti-CTLA-4 antibody). Inhibition of CTL-4 and PD-1 results in downregulation of regulatory T cells and a concurrent upregulation of T effector cells (T-helper cells). As a result, more cytokines are produced. These processes might explain the improvement in PFS and overall survival (OS) in multiple types of malignancy, including cutaneous melanoma (CM) [6]. As their efficacy has been proven in metastatic CM, immune checkpoint inhibitors have been tried in UM and the results are promising [7]. However, efficacy in UM tends to be lower, possibly due to a higher ratio of exhausted T-cells in immune infiltrates of UM metastases compared to CM [8]. Another promising therapeutic option in metastatic UM is Tebentafusp, a bispecific protein that redirects T-cells to attack gp100 expressing cells. After binding to peptide-human leukocyte antigen (HLA) complexes, polyclonal T-cells are recruited and activated for cluster of differentiation (CD) 3-mediated cytokine and cytolytic mediator release. As Tebentafusp redirects T-cells in specific HLA complexes, only patients with HLA-A*02:01 positive patients profit from the experimental therapy. Recently, patients receiving Tebentafusp had an elongated OS and PFS compared to the control group consisting of Pembrolizumab, Ipilumab, or Dacarbazine treatment (median OS: 21.7 months, vs. 16 months, respectively; median PFS: 3.3 months vs. 2.9 months, respectively) [9].

Chromosomal aberrations and gene mutations in UM that correspond with prognosis are well characterized. There are four distinct molecular subclasses, and their respective outcome has been identified [10,11]. The loss of BRCA1-Associated Protein 1 (*BAP1*) along with chromosome 3 loss and chromosome 8q gain is correlated to the worst prognosis, a mutation in splicing factor 3B subunit 1 (*SF3B1*) causes aberrant splicing due to the use of alternative branch points by the spliceosome complex and is associated with an intermediate metastatic risk, whereas Eukaryotic Translation Initiation Factor 1A X-Linked (*EIF1AX*) mutations will generally not result in metastases [12,13,14]. Furthermore, clinical parameters indicative for poor prognosis include higher age, higher mean basal tumor diameter and tumor thickness, extraocular spread, and ciliary body involvement [15].

Although metastatic risk is not influenced by intraocular biopsies [16], they harbor an inherent risk of complications, such as persistent hemorrhage requiring surgical intervention or rhegmatogenous retinal detachment (both in 1% of the fine needle aspiration biopsies (FNABs)). Rarer complications include seeding at the needle tract [17], extra-ocular extension of uveal melanoma [18], and endophthalmitis [19,20]. Furthermore, FNAB will not yield prognostic information in a substantial number of cases. Especially in smaller tumors (<5 mm in height) and posteriorly located tumors is this the case [21]. In addition, worsening of visual acuity (VA) is reported in 13% of the patients who underwent FNAB [21] and the VA worsened in 42% of patients undergoing transvitreal retinochoroidal biopsy [22]. However, worsening of the VA could also be explained by tumor or radiation effects.

Disease progression surveillance without prior genetic knowledge will typically include a 6-monthly liver imaging (either ultrasound or magnetic resonance imaging) and liver function tests for all UM-patients. In the UK, liver screening has been estimated to cost the NHS between £108.000 and £228.000 yearly [23]. Knowledge of such prognostic genetic alterations may, therefore, lead to more individual and cost-effective follow-up strategies.

In summary, the emerging trend in therapeutic options, therapeutic studies, and the reticence regarding invasive biopsies highlight the need for effective and efficient surveillance and non-invasive prognostication. This review addresses liquid biopsies, with an emphasis on analysis of circulating tumor cells (CTCs), circulating tumor DNA (ctDNA), tumor derived exosomes, proteins, and metabolites. Additionally, medical imaging will be discussed in terms of potential prognostic value and clinical use.

## 2. Liquid Biopsy

The term liquid biopsy is a contradiction in terms as biopsies inherently refer to obtaining tissue. Nonetheless, the term beautifully comprises the intention of obtaining tumor information from (the content of) fluids. These blood-based liquid biopsies are minimally invasive, quick, and easy to conduct and undergo. The risks are next to none. When peripheral blood is used for diagnosing or prognostication, usually less than 20 mL is sufficient [24,25,26], which can be withdrawn easily in most, if not all, patients [27].

The possibilities for blood-based tumor biomarkers are vast and encompass, among others, (the cargo of) CTCs, DNA, RNA, exosomes, proteins, and metabolites.

### 2.1. Circulating Tumor Cells

CTCs were first described in 1869 by Dr. Thomas Ashworth, who saw the similarity of cells in the tumor and tumor cells circulating in blood. Since then, CTCs have been of interest in different fields of cancer [28,29]. CTCs are clones of the original tumor and in a heterogenous tumor, multiple, genetically, different CTCs could be found in the bloodstream [30]. Multiple studies have analyzed the genetic composition of CTCs. DNA [26], RNA [31,32], proteins [33], micro RNAs (miRNAs) [34], and metabolites [35] have been found in CTCs, albeit in low quantities.

The mechanism behind intravasation of these tumor cells is not yet fully understood. Two main hypotheses exist. On the one hand, a passive shedding of cells is postulated, where clumps of cells break off the primary tumor and will enter the bloodstream. On the other hand, an active transition of cells to a more mobile state through an epithelial to mesenchymal transition is hypothesized [33]. As such, solitary CTCs as well as clusters are present in blood.

Aceto et al. have shown, in breast and prostate cancer, that single CTCs are more abundant. However, single CTCs do not harbor the same metastatic potential as clusters of CTCs, which are comprised of an aggregation of neighboring cells from the primary tumor (i.e., oligoclonal cluster from primary tumor). These clusters broke off the primary tumor and entered the bloodstream. Mouse models showed a 23–50-fold increase in metastatic potential of CTC clusters compared to single CTCs [28]. These findings translate to humans, as worse OS and PFS were reported in patients with CTC clusters [28,36]. Additionally, it is important to note that the presence of circulating tumor cells do not necessarily represent a metastatic potential [28].

In 1993, over a century after Dr. Ashworth’s discovery of CTCs, Tobal et al., among others [37,38], used reverse transcription PCR (RT-PCR) on the lysed cell pellet of 5 mL peripheral blood to detect transcripts of the tyrosinase gene, a gene encoding a protein crucial for melanin production and in normal conditions not present in blood, in the blood of UM-patients. They detected Tyrosinase transcripts in half of the UM-patients with metastatic disease and a quarter of the patients without metastases [38]. Since then, CTC isolation and characterization has progressed in UM.

#### 2.1.1. CTC Enrichment and Enumeration

CTCs are rare, only a few tumor cells can be found in a single tube of peripheral blood from UM-patients with localized disease [24,26,39,40]. Therefore, sensitive and precise isolation methods are needed.

CTCs can be captured using different techniques, such as filtration-based or immunomagnetic, with the latter being more popular in UM research [26,40,41,42]. A combination of a microfluidic device containing immuno-affinity properties was also developed, in which a chip containing columns covered with epithelial cell adhesion molecule (EpCAM) antibodies can capture CTCs [43]. Filtration-based enrichment is based on the difference in size and compressibility of CTCs and (white) blood cells. Even though the diameter of non-hematological tumor cells varies, the mean diameter of nuclei originating from non-hematological metastases differs substantially from the diameter of lymphocytic nuclei (9.28 µm vs. 4.95 µm (*p* < 0.001), respectively) [44]. Immunomagnetic or affinity-based enrichment relies on the expression of tumor cell specific proteins on the cell membrane of CTCs [45]. Using filtration-based CTC enrichment, Mazzini et al. detected CTCs in 17 out of 31 non-metastatic patients. They showed no correlation between the presence of CTCs and clinical and histopathological prognostic parameters. They did report a significant difference in longest basal diameter, tumor height, PFS, and OS after stratifying patients in groups based on the detection of more versus less than 10 CTCs per 10 mL blood [42].

For capturing CTCs, immunomagnetic isolation is mostly used [24,26,39,40,41,46] in UM. The specificity of antibodies used herein is essential to capturing CTCs. Moreover, different antibody bound magnetic particles can be used for capturing CTCs and they have their own characteristics. Thermo Fisher DynaBeads (Thermo Fisher Scientific, Waltham, Massachusetts, United States) are spherical particles ranging from 1 to 5 µm. Due to their size, they have a relatively strong magnetic power. Dynabeads can be captured without a magnetic mesh or microchip [26]. However, the size of DynaBeads can complicate downstream analyses. Veridex ferrofluids are small, 100 nm-sized, ferroparticles (Fe_3_O_4_) and are labeled with antibodies (EpCAM or melanoma cell adhesion molecule (MCAM)). These ferrofluids are used in the closed, food and drug administration (FDA) approved system CellSearch (Menarini Sillicon Biosystems, Bologna, Italy), which complicates creating custom antibody-bead combinations. However, it is the golden standard method for CTC isolation. The ferrofluids are incubated with a blood sample and the CTC will be drawn to the slide while other cells are discarded [47]. Another type of beads are Miltenyi magnetic-activated cell sorting (MACS) beads. They have a heterogeneous size ranging from 30–100 nm. They do not infer with flowcytometry or other fluorescent applications [48]. For capturing or depletion of bound cells to the magnetic particles, different columns with a magnetized matrix are required.

##### CellSearch

The CellSearch assay uses ferrofluids bound to MCAM to capture melanoma CTCs. Afterwards, they are stained for melanoma-associated chondroitin sulfate proteoglycan (MCSP) (melanoma cells), CD34 (endothelial cells) and CD45 (leukocytes) to differentiate between melanoma CTCs and other cells expressing MCAM. It was first developed for capturing CM cells, where 88% of spiked-in cell line cells was recovered. However, in patients, only in 23% (18/79) of the metastatic CM blood samples had ≥2 CTCs per 7.5 mL blood [29] (Table 1). Other studies have used the same system and protocols for both localized and metastatic UM and CTCs were found more frequently, which could possibly be explained by more robust or abundant CTCs.

In localized disease, 50% of the patients had detectable CTCs, with a median of 2 CTCs per 7.5 mL peripheral blood [40]. Anand et al. detected 1.83 CTCs (mean) in 10 mL peripheral blood in 8 of 20 analyzed patients (Table 1) and the CTC count correlated to worse prognostic factors as monosomy 3 and 8q gain and ultimately correlates independently to a worse OS. Additionally, CTC enumeration was performed in parallel to conventional imaging (computed tomography (CT) or magnetic resonance imaging (MRI) of the abdomen). They had found an increase in CTCs prior to imaging in 60% (3/5) of the patients who developed metastases, in the other two patients who developed metastases no CTCs were recovered [24].

In metastatic disease, the CTC count was higher, 3 (median) CTCs per 7.5 mL, but CTCs were detected in fewer patients (30%) [39] (Table 1). Moreover, Bidard et al. found a correlation between CTC count and tumor volume (R^2^ = 0.28, *p* = 0.005), and subsequently tested ctDNA levels (R^2^ = 0.63, *p* < 0.0001). Surprisingly, two patients with high CTC count (12 and 20 CTCs/7.5 mL) had end-stage disease for which they received palliative care shortly after phlebotomy, but CTC count was not found as an independent prognostic factor [39]. On the other hand, Anand et al. observed a worse OS in patients with detectable CTCs and OS was correlated with CTC count. They reported a mean of 9 CTCs per 10 mL blood in 13 of 19 analyzed patients [24].

##### MACS

Ulmer et al. combined Miltenyi microbeads (Miltenyi Biotec, Bergisch Gladbach, North Rhine-Westphalia, Germany) with MCSP and 2.5 (median) CTCs were found per 50 mL heparinized peripheral blood in 19% of UM-patients with localized disease [49] (Table 1). Using the same protocol, their group analyzed the presence of CTCs before and after primary treatment modalities for UM (enucleation, endoresection, Ruthenium-plaque, stereotactic radiotherapy, and transpupillary thermotherapy), in which they did not observe a difference [50].

##### DynaBeads

In 2014, Tura et al. modified and enhanced the protocol previously published by Cool-Lartigue et al. [52] in which bead-bound antibodies against CD63 and gp100 were used. Using these two markers, they detected CTCs in 93.6% (median: 3.5 cells per 10 mL whole blood) of patients harboring UM without clinically detectable metastases [41]. Beasley et al. first also used MCSP alone and detected CTCs in 69% (18/26) of patients with primary UM. They did not find a correlation between the quantity or presence of CTCs and histopathologic or genetic prognostic parameters [26]. More recently, their follow-up study using a multimarker approach (ATP-binding cassette sub-family B member 5 (ABCB5), gp100, MCAM, and MCSP) detected a median of 3 CTCs in 86% (37/43) of localized cases from 8 mL of blood (Table 1) with the levels of CTCs correlating with shorter PFS and OS [51]. The detection rate had increased by adding melanoma-specific antibodies to their custom assay, which could mean not all UM CTCs express MCSP. The difference in survival characteristics could be explained by these other subtypes of CTCs that were potentially not captured previously, along with an increase in sample size.

When looking at the different bead-types and antibodies used, most studies indicate a worse prognosis when more CTCs are detected.

#### 2.1.2. CTC Characterization

After capturing and isolating individual CTCs, different approaches have been used for characterization of CTCs. Proteins in CTCs can be studied by immunostaining as was shown by Campos et al. who used breast tumor cells that were spiked in and recovered from peripheral blood. The proteins HER2/NEU and TOP2A were identified and quantified and are correlated with Herceptin and Anthracycline therapy response, respectively [53,54]. Showing the clinical relevance of immunostaining and characterizing of CTCs. Other approaches include shallow whole genome sequencing (sWGS), fluorescence in situ hybridization (FISH) and transcriptomics.

#### 2.1.3. CTC Genotyping

Copy Number Variations (CNVs) can be analyzed by whole genome and exome sequencing in CTCs isolated from peripheral blood of small cell lung cancer patients [55]. As specific CNV-profiles correlate to survival, this gives opportunity to minimally invasive risk stratification [14,55]. Sadly, in UM, CTCs are found less abundantly, complicating downstream analyses.

Therefore, low-input sequencing techniques or whole genome amplification is needed [26], which introduces technical artifacts [30]. For example, allelic dropouts after amplification occur regularly [56]. To distinguish true mutations from (whole genome) amplification artifacts, DNA barcodes can be added to the original DNA alleles prior to amplification. More recently an amplification free method has been introduced. Using molecular barcodes on DNA alleles, multiple samples can be pooled for library preparation. Afterwards, genes of interest were sequenced with high concordance to the original tumor [57].

In UM, chromosomal aberrations are correlated to progression free survival [11,12]. Ideally, CNV-profiles derived from CTCs will harbor the same characteristic aberrations as the primary UM tumor and CNV-profiles obtained from CTCs, either using sWGS or FISH, can be used for prognostication [26,46] (Figure 2). Historically, next generation sequencing performed on individual single cells has been difficult. Nonetheless, several studies were successfully conducted, giving insight in CTC biology.

In 2012, single CTC transcriptomics was performed with Switching Mechanism at 5′ end of RNA Template (SMART)-seq, using template switching. This improved the full-length coverage (from 3′ to 5′) to 40%, compared to older techniques and coverage of all expressed genes came to 25% from 10 pg of starting material [58]. Using a novel microfluidic chip to minimize cell loss prior to sequencing, Shi et al. have improved coverage and retrieved adequate expression data using SMART-seq2 library preparation, but they needed a minimum of 10 spiked-in CTCs [59]. Interestingly, Gkountela et al. have additionally discovered different cell states in single circulating breast cancer cells compared to CTC-clusters, which have a bigger metastatic potential. They also used SMART-seq2 library preparation on pooled CTCs and CTC-clusters and describe an upregulation of cell–cell junction components and a higher proliferation rate in CTC-clusters [36].

From the content of just over one cell, Lang et al. have successfully produced end-to-end RNA libraries, using Ovation single cell RNA-seq. They concurrently stratified the primary tumor (using the prediction analysis for microarrays for breast carcinoma signatures: Nanostring PAM50 (Nanostring, Seattle, WA, USA)). The expression profiles in these single CTCs were highly discordant to the matched PAM50 signatures of the primary tumor, suggesting that they cannot be used for noninvasive prognostication and the authors explained the discordance as a difference in biologic processes present in a CTC compared to cells in their tumor environment [60].

Epigenetic regulation is a big factor in regulation of expression, which can also be explored in CTCs using whole genome bisulfite sequencing. This transforms unmethylated cytosines into uracil prior to sequencing and are ultimately read as thymines, whereas methylated regions retain their cytosines. Afterwards, methylated and unmethylated regions can be elucidated. Gkountela et al. have uncovered an enrichment of stemness related transcription factors regulating proliferation and pluripotency in CTC-clusters, compared to single CTCs, corresponding to their findings in RNA expression [36].

### 2.2. Circulating Tumour DNA

Cell-free DNA (cfDNA) circulates through the body and is either actively secreted by cells or is the result of cell death [61]. These physiological processes are naturally occurring [62] and cfDNA levels rise after pathologies, such as stroke, auto-immune disease, trauma, myocardial infarction, or cancer [63]. In cancer, ctDNA is released by tumor cells through apoptosis, necrosis, or active secretion [61]. Subsequently, mutations of the primary tumor can be retrieved in ctDNA [26,64]. However, the ctDNA fraction is low and accounts for <0.1–10% of cfDNA in a patient harboring a malignancy [65]. As such, the low fraction of ctDNA can make detection of the driver gene mutations or copy number variations difficult [66].

Circulating DNA is fragmented; cfDNA-fragments are usually between 120–220 bp long. This coincides with the length of a DNA strand wrapped around a nucleosome plus 20 bp of linker DNA bound to histone H1 [67]. ctDNA is more fragmented, with an enrichment of 90–150 bp ctDNA fragments, giving opportunity for in-silico or in-vitro enrichment of ctDNA [25].

Studies have shown a rise in cfDNA in patients harboring a malignancy, which could add prognostic value [68,69]. However, physiological processes could equally elevate cfDNA levels [68].

#### 2.2.1. Clinical Use of ctDNA in UM

In UM, primary driver mutations are activating and mutually exclusive. In over 95% of the cases, somatic hotspot-mutations occur in G-protein α subunits Q and 11 (*GNAQ* and *GNA11*, respectively), cysteinyl Leukotriene Receptor 2 (*CYSLTR2*) or phospholipase C beta 4 (*PLCB4*) [3,70,71]. Due to homogeneity in primary driver mutations, single mutation assays, albeit multiplexed, are efficient and feasible for disease detection and monitoring [64,70]. Tumor size, disease burden and ctDNA levels in primary and metastatic UM have been correlated to ctDNA levels of mutated primary driver genes [70,72].

Of the secondary driver genes (*BAP1*, *SF3B1* and *EIF1AX*), only mutations in *SF3B1* are suited for hotspot-mutation analysis. *SF3B1* mutations result commonly in a missense replacing the arginine at position 625 [11,12,73]. Mutations in *BAP1*, which comprises 50% of UM tumors, are loss of function mutations. They are found throughout the gene and can even encompass the deletion of one or more exons of the BAP1 gene [12]. *EIF1AX* mutations are, as the *SF3B1* mutations, gain or change of function mutations. However, these mutations occur not at a single amino acid, but are spread out five to ten amino acids of the N-terminal region of the protein [14]. This results in only a possibility for *SF3B1* mutations to be found using single hotspot-mutation analysis. Considering the prevalence of *SF3B1* mutations (*SF3B1* mutations occur in 24% of UM-tumors [12]), in about a quarter of the UM-patients, the secondary driver mutation could be uncovered. When multiple techniques are combined in which either *BAP1* or *EIF1AX* mutations can be discovered, mutation analysis for *SF3B1* mutations can be a valuable addition.

UMs are sometimes difficult to distinguish from naevi. Especially when risk factors for potential malignant transition are present. These risk factors include the presence of subretinal fluid, orange pigment, tumor thickness >2 mm, peripapillary location, visual symptoms, and ultrasonographic hollowness [74]. An overlap between naevi with risk factors for malignant transition and small UM-tumors exist and make it hard to distinguish these groups. Moreover, naevi can harbor the same primary driver mutations in *GNAQ* or *GNA11* as UMs do, without malignant transformation. Elevated levels of mutated cfDNA are highly correlated to malignancy and clinical risk factors for UM [75]. Therefore, circulating DNA harboring primary driver mutations could provide information of malignant transformation of these lesions and can be used for follow-up of naevi at risk.

Treatment effects can be evaluated and monitored by analyzing ctDNA. For example, protein kinase C inhibitors have been tested in metastatic UM and treatment response correlated with lower ctDNA at endpoint [70]. Beasley et al. 2018 found a rise in ctDNA prior to radiologic confirmation of metastases in standard patient care [26]. Additionally, Le Guin et al. analyzed cfDNA in blood samples from 135 patients of which 19 developed metastases and two a local recurrence. In 106 patients, no ctDNA was detected at any time. From the patients with disease progression, 17 of 21 patients had detectable ctDNA >5 months after treatment (irradiation) and a spike in ctDNA was observed two to ten months prior to radiologic confirmation. Moreover, the presence of ctDNA was linked to disease progression with 80% sensitivity and 96% specificity [76], suggesting earlier and less burdensome detection of metastases and a potential ability of earlier treatment.

#### 2.2.2. ctDNA Genotyping

Besides disease monitoring, ctDNA could be used for minimally invasive CNV-profiling and, thus, in the case of UM, provide valuable prognostic information, as the patients’ prognosis is highly concordant to certain chromosomal aberrations in UM.

As stated earlier, ctDNA fragments are smaller than cfDNA. The fragmentation can be used for enrichment of ctDNA. Mouliere et al. have evaluated in silico size selection (selecting fragments of 90–150 bp after paired-end sequencing) and in-vitro size selection (using a microfluidic device), in which the in-vitro selection provided minimally invasive CNV-profiles from ctDNA in patients with ovarian cancer after sWGS [25]. Different malignancies can provide varying proportions of short (ctDNA) fragments in cfDNA [25,77]. Thus, the fragment size distribution of ctDNA in blood of UM-patients should be elucidated for enrichment prior to genotyping. In metastatic breast and prostate cancer, CNV-profiles were also obtained from ctDNA, where ultra sWGS, with a genomic coverage of 0.1×, was performed on three to 20 ng cfDNA. CNV-profiles from ctDNA were highly concordant to metastatic laesions (Spearman ρ = 0.763 [78]) from a minimum tumor fraction of 10% [78,79]. However, 10% tumor fraction in blood is not feasible for UM confined to the eye [26,70,76], meaning further enrichment of short fragments is necessary in UM.

### 2.3. Extracellular Vesicles

Extracellular vesicles (EVs) are nano-sized particles secreted by both healthy and pathological cells (i.e., cancer cells), comprising of microvesicles and exosomes. They are being used for intercellular communication [80]. Microvesicles (or ectosomes) originate from the plasma membrane and are formed by outward budding and fission, followed by release of these vesicles. The diameter ranges from 50 to 1000 nm [81]. Microvesicles can contain DNA, messenger RNA (mRNA), miRNA, and proteins [82,83,84]. EVs are involved in cell–cell signaling and modulation of the extracellular environment [81,84]. Moreover, EVs derived from tumor cell lines can drive an oncogenic shift in which proliferation is upregulated after invasion of the target cell. For example, Tsering et al. observed tumor growth after treating BRCA1-deficient fibroblasts with UM-derived EVs in mice [84].

Furthermore, EVs shed by tumor cells have enriched proteomic profiles involved in endocytosis, cell–cell signaling and focal adhesion and metastatic niche formation in UM [84]. Specific upregulated proteins in EVs originating from UM cells include Vimentin [84], melanoma-associated antigens D1 and D2 (MAGED1 & 2) [84], and melanoma antigen (MLANA) [84]. For minimally invasive prognostication, the focus should shift towards characterization of tumor derived microvesicles in patients’ serum.

### 2.4. Exosomes

Exosomes are discoid vesicles originating from endosomes and they are 30–150 nm in diameter. Some debate exists about the DNA load of exosomes. Some may argue that exosomes contain double stranded DNA (dsDNA). Thakur et al. uncovered dsDNA in a small subset of murine melanoma exosomes. Whole genome sequencing of exosomal dsDNA gave an unbiased coverage of the entire genome [85]. Takahashi et al. have also created unbiased CNV-profiles and even reported a homeostatic DNA secretion function for exosomes, in which exosome secretion inhibits aberrant immune response by secreting excess cytoplasmic DNA [86]. On the other hand, Jeppesen et al. were unable to detect dsDNA in exosomes. Concurrently, no DNA binding histones were found within (CD63 or CD83 positive) exosomes. As such, they concluded that DNA must be present on the vesicle membrane [87]. The debate whether exosomal DNA is on the membrane or within the vesicles is not settled. Despite this uncertainty, adequate CNV-profiles have been obtained using exosomal DNA and they could be used as a future biomarker.

Less debate exists about exosomes containing proteins, miRNA, and mRNA [80,87,88]. Several isolation techniques have been developed; however, no technique can purely isolate exosomes [89]. Capture techniques include: ultracentrifugation [80,90], (bead-bound) immuno-affinity [87], and size exclusion chromatography [91]. In UM, ultracentrifugation is mostly used and hereafter exosomes are commonly gold-coated and visualized using scanning electron microscopy [88]. Afterwards, exosomes are verified by expression of endosomal proteins programmed cell death 6-interacting protein (ALIX), Lysosome-associated membrane glycoprotein 1 (LAMP1) and CD63 [88]. Inflammation related proteomic profiles of exosome cargo have been analyzed by isolating exosomes via ultracentrifugation. Specific interleukins (IL-2, IL-11, IL12(p40) and IL-27) were upregulated in metastatic UM, compared to primary UM. These interleukins are involved in both pro and anti-tumor response. IFN-γ was also found upregulated in exosomes derived from metastatic disease, which is correlated to metastases and overall more extensive disease [92]. As IFN-γ was highly upregulated in only the metastatic patients, this is potentially an exosomal marker for monitoring disease progression.

UM derived exosomes have not yet been studied elaborately. Eldh et al. [88] isolated exosomes directly from the liver circulation in metastatic UM-patients. During isolated liver perfusion, perfusate was collected and exosomes were pelleted by ultracentrifugation. In addition to a higher quantity of exosomes, distinct miRNA patterns were found in patients versus healthy controls. MiRNAs are small RNA molecules that bind to the 3′ untranslated regions of mRNA and inhibit translation of these mRNAs and, therefore, regulate protein production [93]. For miRNA expression, the brain cancer related miRNA panel on the RT^2^ miRNA PCR array (Qiagen) is used. Upregulated miRNAs include miR-107, -124, -210, -320a, -370, and -486-5p [88]. miR-21, -34a, and -146a were found in exosomes captured from vitreous humor and peripheral blood and validated in UM-tissue [94].

MiRNAs provide valuable insights into the epigenetic regulation by exosomes. However, both studies only addressed upregulation between healthy individuals and UM-patients. Subsequently, no differentiation can be made between the low and high-risk UM-patients. Moreover, as the miRNA patterns do not overlap between both studies, uncertainty rises for the ability of these miRNAs as biomarkers.

### 2.5. Micro RNA

Circulating miRs were also isolated and sequenced directly from plasma and differentially expressed miRs could be used as a noninvasive prognostic tool for detection of (high-risk) UM. Furthermore, serum miRNAs were proposed as a surrogate marker for CTCs in breast cancer as capturing and isolating CTCs can be difficult because certain miRNAs were upregulated in CTC-positive patients with breast cancer [95].

In tumor tissue, the concentration of tumor derived miRNAs is higher than in peripheral blood. Smit et al. [96] have investigated miRNA expression using an Ion Proton sequencer (Life technologies) in high-risk tumors, consisting of *BAP1*-mutated tumors and absent BAP1 expression, intermediate-risk tumors, consisting of *SF3B1*-mutated tumors, and low-risk tumors, consisting of *EIF1AX*-mutated tumors. They reported an upregulation of miR-16-5p, -17-5p, -21-5p, -132-5p, and -151a-3p in high-risk tumors, while miR-99a-3p, -101-3p, -181a-2-3p, -181-5p, -378d, -1537-3p, and miRNA precursor let-7c-5p were downregulated in these high-risk tumors [96].

Multiple groups have identified differential expression in miRNAs using different techniques from plasma isolated from peripheral blood. Using quantitative RT-PCR, 6 monosomy 3 UM-patients were followed and miRs were analyzed at of disease and metastasis. At time of metastasis miR-20a, -125b, -146a, -155, and -223 were upregulated and miR-181a was downregulated [97]. Triozzi et al. [98] analyzed the expression of 674 miRs in tumor tissue and plasma between monosomy and disomy 3 UM-patients using the HTG quantitative nuclease protection assay (qNPA). One upregulated miR in tumor tissue was also found upregulated in plasma: miR-92b (*p* < 0.02) [98]. Using a Taqman low density array (TLDA) including 754 miRNAs, MiR-181a was found upregulated in plasma, while downregulated in tumor tissue. MiR-21, found upregulated in exosomes of UM-patients compared to healthy controls [94] and upregulated in tumor tissue of high-risk UM compared to low and intermediate-risk tumors [96], was not differentially expressed in plasma of monosomy and disomy 3 patients [98]. MiR-20a was not found differentially expressed by Triozzi et al., when Achberger et al. found an increase at time of metastasis. Furthermore, Achberger found a downregulation of miR-181a in metastatic disease, while Triozzi et al. found an upregulation in monosomy 3. Whereas in tumor tissue of high-risk UM miR-181-5p and miR-181a-2-3p were both downregulated [96]. Both studies show an upregulation of miR-223 in plasma of high-risk or metastatic patients. Ragusa et al. reported an increase of miR-618 in UM-patient derived exosomes from vitreous humor and plasma isolated from peripheral blood, while miRNAs isolated directly from vitreous humor showed a decrease of miR-618 in UM-patients versus healthy controls [94]. Table 2 shows a summary of the blood-derived miRNAs, their involvement in (uveal) melanoma and its up or downregulation in UM, high-risk or metastatic patients compared to either healthy individuals, low metastatic risk, or non-metastatic patients.

The advantages of miRs as biomarkers encompass a long half-life, due to in part the incorporation in EVs, easy accessibility through liquid biopsies, and high specificity and sensitivity [99,100,101,102]. miRNAs can have opposite functions based on the context and disease in which they are studied, which makes interpreting study results difficult. Furthermore, studies make use of miRNA panels, hybridization-based arrays as well as (next generation) sequencing technologies. These techniques all have differences in analyzed miRNAs and sensitivity thereof. Differentially expressed miRNAs could provide prognostic information, but the differences in disease state, or context, and used techniques make cross-study outcome statistical comparison and replication challenging.

In the context of UM, miRNAs should show concordant expression between cohorts with the same disease state; however, expression patterns do not overlap between monosomy 3 patients (i.e., high-risk of metastasis) and already metastasized tumors. The true reasons for the distinction in miRNA between high-risk localized and already metastasized UM must be elucidated, before making miRNA expression a clinically viable minimally invasive biomarker.
biomedicines-10-00506-t002_Table 2Table 2Regulatory effects in cutaneous or uveal melanoma of upregulated microRNA found in plasma and extracellular vesicles in vitreous humor, peripheral blood, blood collected during isolated liver perfusion, and/or UM-tissue. Expression compared to: UM-patients versus healthy controls; Metastatic patients versus localized disease; Monosomy 3 versus disomy 3. Abbreviations: DE: differentially expressed; RT-PCR: reverse transcription polymerase chain reaction; EVs: extracellular vesicles; TLDA: Taqman low density array; RT^2^ array: Qiagen RT^2^ miRNA PCR array (brain cancer panel); qNPA: HTG molecular diagnostics quantitative nuclease protection assay; VH: vitreous humor.microRNAExpressionFound in Tissue TypeSequencing TechniqueModulatory EffectmiR-20a [97]Upregulatedin UM-patients and metastatic patientsPlasmaRT-PCRPromotes cell proliferation and migration by modulation of the cell cycle, focal adhesion and phosphoinositide 3-kinase (PI3K)-AKT signaling pathway [103,104].miR-20a [98]Not DE between monosomy and disomy 3PlasmaRT-PCRmiR-21 [94]Upregulatedin UM-patientsEVs: vitreous and FFPE UM-TissueTLDAPromotes tumor growth, invasion, and metastasis, by regulation of tumorsuppressors (p53) [105] in CM and UM [106,107,108].miR-21 [98]Not DE between monosomy and disomy 3PlasmaRT-PCRmiR-34a [94]Upregulatedin UM-patientsEVs: vitreous and FFPE UM-TissueTLDAPDL-1 is regulated by p53 via miR-34, causing immune evasion: UL16-binding protein 2 (ULBP2) is downregulated causing a diminished cell recognition by NK-cells [109,110].miR-92b [98]Upregulated in monosomy 3PlasmaRT-PCRPromotes proliferation and migration in hepatocellular carcinoma. No mechanistic information is known in (U)M [111].miR-107 [88]Upregulatedin UM-patientsEVs: isolated liver perfusateRT^2^ arrayInhibits cell proliferation, migration, and invasion in CM. Highest expression is seen in metastatic melanoma [112].miR-124 [88]Upregulatedin UM-patientsEVs: isolated liver perfusateRT^2^ arrayHomeobox 11 (HOXA11)-antisense RNA promotes proliferation and invasion by inhibiting miR-124 in UM [113]. MiR-124 inhibits proliferation, migration, invasion and promotes apoptosis of melanoma cells [114].miR-125b [97]Upregulated in metastatic diseasePlasmaRT-PCRInduces apoptosis and inhibits proliferation and migration of CM cell line cells by targeting neural cell adhesion molecules (NCAM) [115].miR-146a [94,97]Upregulatedin UM-patientsEVs: vitreous, plasma and FFPE UM-Tissue; PlasmaTLDA/RT-PCRMiR-146 has a potential immunosuppressive role, when upregulated it causes NK-cell proliferation inhibition and apoptosis induction [103]. Additionally, miR-146 is regulated by microphtalmia-associated transcription factor (MITF) [94].miR-146a [97]Upregulated in metastatic patientsPlasmaRT-PCRmiR-155 [97]Upregulatedin metastatic patientsPlasmaRT-PCRIs upregulated in UM-tumors and promotes invasion and proliferation by targeting Nedd4-family interactive protein 1 (NDFIP1). NDFIP1 is necessary for ubiquitination and translocation of, tumor suppressor, PTEN [116,117,118]. Upregulation is correlated to monosomy 3 status [118].miR-155 [94]Downregulated in UM-patientsVH and VH EVsTLDAmiR-181a [97]Downregulatedin metastatic patientsPlasmaRT-PCRUpregulation inhibits CTD small phosphatase like (CTDSPL) expression, which in turn promotes cell cycle progression in UM cells [119].miR-181a [94]Downregulated in UM-patientsVH and VH EVsTLDAmiR-181a [97]Upregulated in UM-patientsPlasmaRT-PCRmiR-210 [88]Upregulatedin UM-patientsEVs: isolated liver perfusateRT^2^ arrayTargets vascular endothelial growth factor (VEGF)-dependent endothelial cell migration and tube formation factor ephrin A3 and subsequently promotes angiogenesis by formation of capillary like structures [120] and is induced by hypoxia in melanoma [121].miR-223 [98]Upregulated in monosomy 3PlasmaRT-PCR/qNPARegulates and suppresses myeloid derived suppressor cells, which expand during pathology and are related to UM [122,123,124].miR-223 [97]Upregulated in UM and metastatic patientsPlasmaRT-PCRmiR-320a [88]Upregulatedin UM-patientsEVs: isolated liver perfusateRT^2^ arrayInhibits the epithelial to mesenchymal transition (EMT) by regulating the transforming growth factor (TGF)-β1/suppressor of mothers against decapentaplegic (SMAD) pathway [125,126].miR-370 [88]Upregulatedin UM-patientsEVs: isolated liver perfusateRT^2^ arrayOverexpression promotes cell growth and invasion of melanoma cells by regulation of pyruvate dehydrogenase E1 subunit Beta (PDHB) [127].miR-486a-5p [88]Upregulatedin UM-patientsEVs: isolated liver perfusateRT^2^ arrayOverexpression inhibits proliferation and migration in hepatocellular [128] and colorectal cancer [129]; however, no mechanistic information is available for (U)M.


### 2.6. Proteins

In 2007, Pardo et al. suggested proteomics for biomarker discovery. Using mass spectrometry they saw a higher abundance of melanoma-specific gp100 and Cathepsin D in sera from UM-patients compared to healthy individuals [130,131]. Ultimately, when looking for prognostic biomarkers retrieved from peripheral blood for prognostication, a distinction should be made between high and low risk of metastases.

Angi et al. found 15 proteins upregulated in the secreted proteins of high-risk UM-patients, among which were neurosecretory protein VGF, protein V homolog (V), thrombospondin-2 (THBS2), neuropilin-2 (NRP2), peptidyl-glycine alpha-amidating monooxygenase (PAM), serine protease HTRA1 (HTRA1), plasminogen activator inhibitor 1 (SERPINE1), Laminin subunit alpha-1 (LAMA1), connective tissue growth factor (CTGF), extracellular matrix protein 1 (ECM1), and Insulin-like growth factor-binding protein 7 (IGFBP1) [132]. Moreover, these highly upregulated proteins of presumed exosomal origin are involved in extracellular matrix remodeling and cancer cell migration and invasion [132,133].

Protein abundance can be used for monitoring disease progression; soluble c-Met correlates to metastatic disease in cell lines, xenograft mice, and peripheral blood of UM-patients [134]. Another protein that is reported to be overexpressed in peripheral blood of patients harboring metastatic UM is Parkinson disease protein 7 (DJ-1), which has a potential anti-apoptotic function [135]. Additionally, elevated levels of osteopontin, S100 calcium binding protein beta (S-100β), and melanoma inhibitory activity (MIA) have been reported in metastatic UM. Multiplexed analysis combining these three proteins results in high sensitivity and specificity (area under the curve = 0.91) for the detection of hepatic metastases [136].

More recently, a multiplex assay of heat shock protein 27 and Osteopontin can accurately differentiate between metastatic and non-metastastatic UM after initial treatment [137]. Strikingly, no overlap is seen in proteins upregulated in metastatic UM between the different studies.

### 2.7. Metabolites

Tumor metabolomics has been performed for several years, which show an upregulated energy metabolism by utilizing the oxidative phosphorylation (OXPHOS) and succinate dehydrogenase (SDHA) pathways in UM [138,139]. However, these studies have been performed on tumor tissue and are not useful for minimally invasive prognostication yet.

Recent advances in machine learning opens opportunities for metabolic pattern recognition in peripheral blood samples. As a result, indirect tumor effects can be defined and these patterns can be recognized, as shown by Huang et al. for lung adenocarcinoma [140]. No such metabolomic profiles have been retrieved from peripheral blood of UM-patients but as for lung cancer, the detection of these in serum samples might hold prognostic value in the future.

### 2.8. Hepatic Biomarkers

Historically, hepatic biomarkers are used for monitoring metastasis formation to the most commonly affected organ [23]. Especially higher serum gamma glutamyl transferase (γGT) and serum lactate dehydrogenase (LDH) levels are correlated to metastases [141]. However, standard surveillance: physical examination, abdominal ultrasonography (US), chest X-rays, and liver function tests are not adequate for early detection of metastases [142,143]. Moreover, no prognostic information can be ascertained from the liver function tests at this point.

## 3. Prognostic and Clinical Use of Medical Imaging

Medical imaging plays a fundamental role in the initial screening, staging, and diagnosing of UM [13]. Currently, the diagnostic procedure entails clinical examination with slit lamp and indirect ophthalmoscopy and US. To objectify tumor thickness, orange pigment and subretinal fluid easier optical coherence tomography (OCT) and fundus autofluorescence imaging can be performed [144]. For visualizing the intrinsic tumor vasculature, fluorescein angiography and indocyanine green can be used [145]. However, at this moment of time these modalities do not offer added prognostic information.

After initial staging, the gold standard for metastasis surveillance is imaging, common laboratory tests and liver function tests [7]. It is common practice to screen UM-patients every 6–12 months by abdominal imaging, depending on tumor risk stratification [13]. Initial staging can be performed by (in ascending order based on resolution, cost, and patient burden): US, CT, MRI, or positron emission tomography/CT (PET/CT).

### 3.1. Ultrasonography

Conventional A- (amplitude, one dimensional) and B- (brightness, two dimensional) scans use ultrasonic waves to distinguish between tissue types through differences in density and elasticity. As such, a tumor can be detected by different uptake and reflection of these sound waves compared to healthy tissue [146]. US shows a typical homogenous gray pattern. The amplitude of an A-scan could be used to differentiate between cell types in the tumor; however, it cannot be used for reliable subclassification in UM. US can provide easy visualization of the tumor, without much burden for the patient. Therefore, it can easily be used for follow-up of the primary tumor when the eye is preserved after treatment.

In 2004, Coleman et al. reported an estimated prospective performance of 80.1% when they used ultrasound to evaluate extracellular matrix loops by backscatter analysis [146]. These extracellular matrix loops in the tumor, either individual or network forming loops, are correlated to a worse prognosis [13,147].

### 3.2. Optical Coherence Tomography

OCT can be used primarily for retinal assessment by providing cross-sectional images based on optical backscatter patterns [148,149]. In recent years, advances have been made. The addition of enhanced depth imaging (EDI) makes a deeper visualization of the choroid possible [150]. Blood vessels can be visualized by computationally measuring blood flow, thus removing the need for an intravenously administered fluorescent dye [151].

Using OCT, EDI, and OCT-angiography; the tumor and its vasculature can be visualized and can help differentiate between melanoma and other tumors. Especially by observing a highly irregular and dense vasculature in the outer retinal layer and choroid capillary layers [150,152]. Furthermore, Li et al. observed a widening of the deep foveal avascular zone, and the capillary vessel density of both deep and superficial vessels was reduced [153].

Swept-source OCT uses a longer wavelength laser, which allows for increased penetrance and deeper imaging. Therefore, it could be used as a non-invasive diagnostic tool to differentiate between UM and naevi, as choroidal vessels are deeper in UM [154]. However, Pellegrini et al. performed swept-source OCT angiography and visualization was difficult due to pigmentation and subretinal fluid, both present in most UM cases [155].

Some differences in OCT images are documented between naevi and UM. Besides initial examination of the primary tumor, OCT could be used as a means for follow-up of the primary tumor in small and shallow UM. Above all, due to its ease of use and widespread availability. Unfortunately, no prognostic information for UM can be ascertained using OCT.

### 3.3. Magnetic Resonance Imaging

MRI is not part of standard imaging of the primary tumor. This is mainly due to practicality and historic difficulties in the duration of imaging (the eye needs to be completely still during MRI), and spatial resolution combined with artefacts caused by the air-to-bone border [156,157]. Although it is difficult to circumscribe flat shaped tumors using MRI, it can be used for assessing the tumor volume and visualizing extraocular spread [158], which are both independent prognostic factors for PFS and OS [13]. Ferreira et al. described extrascleral extension using MRI in three cases, which were later confirmed histopathologically or visually during surgery. Using MRI, one case of extrascleral extension was missed. They were also able to determine nerve sheet invasion, which was confirmed by the ophthalmic pathologist [147].

Recent studies have explored the prognostic capabilities of MRI in UM [147,156,157,158,159]. Diffusion-weighted (DW) MRI differentiates between tissue cellularity by evaluating water diffusion. The quantitative value of DW imaging, the apparent diffusion coefficient (ADC), can be used to assess therapy-induced regression of the tumor [159]. US is the gold standard for local tumor staging and surveillance and usually tumor regression is evaluated by US [13]. When ADC is measured, the treatment response can be detected earlier, as the tumor volume, which is used for US follow-up, decreases slowly after irradiation. Furthermore, in tumors with a low ADC irradiation tends to perform better than in high ADC tumors [158,159], which could be used in determining the best treatment strategy in the future. ADC is inversely correlated to tumor prominence [147], which could also be obtained by conventional imaging. Therefore, at present ADC does not add prognostic information.

Dynamic contrast-enhanced MRI (DCE-MRI) can visualize the microvascular blood flow of the tumor by monitoring the response of tissue on a paramagnetic contrast agent [160]. Wei et al. have shown a significant decrease in K^trans^, a measure for capillary permeability, in metastatic UM. These results suggest that tumors capable of metastasizing produce more but low-quality vessels in the tumor. Ultimately, DCE-MRI could possibly be of added prognostic value [157]. Overall, MRI could provide information previously only obtainable by histopathology.

### 3.4. Computed Tomography

#### 3.4.1. Computed Tomography

CT generates cross-sectional images using Röntgen radiation, possibly combined with contrast agent. CT-scans are frequently used for metastatic detection or confirmation, preceded by US and conventional hepatic biomarkers [161].

Hepatic metastases can arise as solitary or multiple lesions. A miliary metastatic pattern is also reported, which consists of numerous small metastases throughout the liver. The number of metastatic lesions seen on CT or MRI correlate to OS, with more metastases correlating to shorter OS and OS is comparable in patients who have a miliary metastatic pattern or more than 10 metastatic lesions [162]. However, Yavuzyigitoglu et al. reported that gene mutation status, the most important factor for survival, does not correlate with the metastatic pattern [162]. So far, CT is only used for metastatic detection and no prognostically important information is derived.

#### 3.4.2. Positron Emission Tomography/Computed Tomography

PET/CT visualizes the uptake of radiolabeled molecules by tumor cells. In UM, ^18^F-FDG is used for measuring metabolic activity of tumor cells, either in the primary tumor or metastases [163,164].

Several studies show correlation of PET/CT uptake with prognostic variables; an increased uptake in tumor cells with chromosome 3 loss [163,164], increasing tumor size [163] and AJCC T-classification [163] are reported. Nonetheless, no association with chromosome 8q gain, another indicator for prognosis, has been reported [163].

For detection of metastases, FDG-PET/CT is not undisputed, as multiple studies show a lack of uptake in some of the metastatic lesions [165,166].

## 4. Clinical Relevance and Added Value of Combining Minimally Invasive Modalities

The aforementioned modalities all have limited prognostic value and, when using a technique alone, the added value is currently often insufficient. In UM, the tumor burden is low and blood-based biomarkers are usually found in lower quantities compared to, for example, lung [55], breast [78,167], or prostate cancer [31,168,169]. Therefore, the use of CTCs, ctDNA, or other biomarkers is usually hampered in UM-patients. However, these biomarkers combined with imaging might provide enough prognostic information.

Conventional genetic and histopathologic features can estimate PFS with a concordance of ±0.8 [11]. By combining multiple noninvasive modalities, the same precision needs to be accomplished. An advantage is that multiple blood-based biomarkers can be extracted from a single tube; after centrifugation, blood is separated in layers, with ctDNA and metabolites remaining in the plasma layer and CTCs can be isolated from the white blood cell layer. Liquid biopsies can also be taken from aqueous and vitreous humor. However, these techniques are more invasive and harbor risk in respect to the eye and they were not discussed extensively in this review.

Circulating tumor cells possess genetic information from the primary tumor, which consists of DNA, RNA, miRNA, proteins, and metabolites. Obviously, valuable information that can be used for a better understanding in the metastatic process and can possibly provide new insights for novel therapeutic options. CTCs can be genotyped, phenotyped, and enumerated. Therefore, CTCs harbor a valuable opportunity for prognostication. Unfortunately, CTCs are laborious, expensive, and time-consuming to capture and analyze. Therefore, CTC analysis is best suited for initial prognostication. For disease monitoring, other techniques will be more suited.

For example, ctDNA-analyses can be cheaper, and are easier and faster. ctDNA levels can be eloquently analyzed. The downside of analyzing ctDNA levels by droplet digital PCR (ddPCR) is that the targeted mutation needs to be known and sequencing of CTCs could provide such data. Currently, when no tissue is available for mutational analysis, hotspot-mutations in primary driver genes and *SF3B1* could still be analyzed using ddPCR. Afterwards, during follow-up visits, disease progression can be monitored faster and more accurate by using ddPCR than by conventional follow-up. Furthermore, when the amount of ctDNA is sufficient, CNV-profiles can be obtained using sWGS and provide additional minimally invasive prognostic information.

During initial staging, multiple imaging modalities are used. By adjusting these procedures, some prognostic information can be ascertained and in combination with liquid biopsies this could be of added value for stratification of high and low-risk patients. Furthermore, both medical imaging and computer science have progressed in recent years. These advancements make pattern recognition by machine learning nowadays a valid option to find new patterns and correlations between images and survival or metastatic risk.

Ultimately, more attention should be given to combining these techniques for clinically beneficial studies in the near future. Depending on the robustness of the minimally invasive diagnostic tools, one or more techniques could be used for prognostication and validation thereof.

## Figures and Tables

**Figure 1 biomedicines-10-00506-f001:**
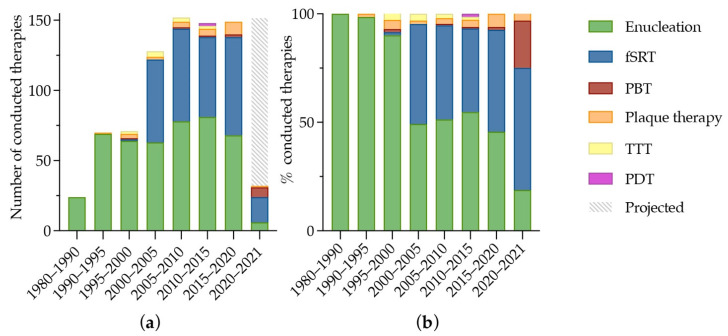
Overview of 774 UM-patients treated at the Erasmus MC and Rotterdam Eye Hospital, Rotterdam, The Netherlands from 1980 to 2021. At the Erasmus Medical Center, fractionated stereotactic radiotherapy (fSRT) was performed since December 1999 and proton beam therapy was available from January 2020. Plaque therapy was performed at the Leiden University Medical Center since January 1990. (**a**) the number and type of therapy conducted; and (**b**) the respective fraction of therapy type per time period. In 2020–2021, 24 patients were treated for UM. The gray and white striped bar marks the expected number of UMs in the Rotterdam Ocular Melanoma Study group (ROMS)-cohort from 2021 to 2025. Abbreviations: fSRT: fractionated stereotactic radiotherapy; PBT: proton beam therapy; TTT: transpupillary thermo therapy; PDT: photodynamic therapy.

**Figure 2 biomedicines-10-00506-f002:**
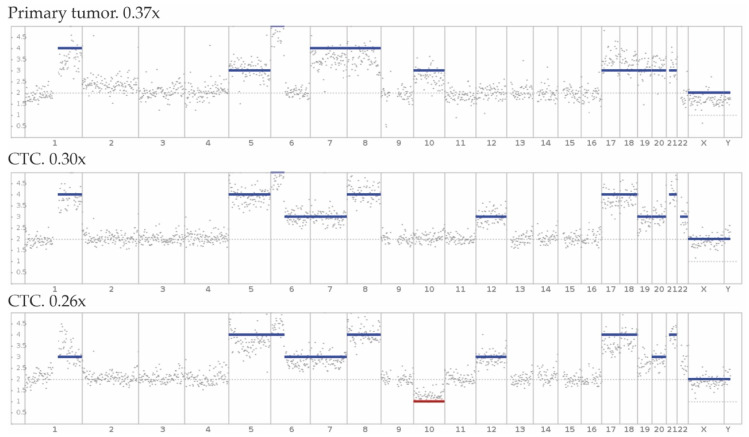
Modified from Beasley et al. [26] showing CNV-profiles derived from the primary tumor (FFPE tissue) and CTCs isolated from 10 mL peripheral blood of the same patient after whole genome amplification and shallow Whole Genome Sequencing (genomic coverage: 0.37×, 0.30× and 0.26×, respectively).

**Table 1 biomedicines-10-00506-t001:** Summary of CTC yield per bead-bound antibody (combination) used for isolating CTCs from peripheral blood in patients with either localized or metastatic UM. Median marked with an * is calculated from available data. Abbreviations: MCSP: melanoma-associated chrondroitin sulfate proteoglycan; ABCB5: ATP-binding cassette sub-family B member 5; gp100: glycoprotein 100; MCAM: melanoma cell adhesion molecule.

Bead-Bound Antibody Used	Disease Status	CTC Count Median (Range)	Detection Rate	
			n.	%
MCSP	Localized	2.5 (1–5)/50 mL	10/52	19% [49]
	Localized	1 (1–8)/50 mL	13/94	14% [50]
	Localized	2 (1–37)/8 mL *	18/26	69% [26]
ABCB5, gp100, MCAM, MCSP	Localized	3 (1–89)/8 mL	37/43	86% [51]
CD63 and gp100	Localized	3.5 (1–10)/10 mL	29/31	94% [41]
MCAM (CellSearch)	Localized	2 (1–3)/7.5 mL	4/8	50% [40]
	Localized	1.5 (1–3)/10 mL *	8/20	40% [24]
	Metastatic	2 (1–38)/10 mL *	13/19	68% [24]
	Metastatic	3 (1–20)/7.5 mL *	12/40	30% [39]

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
