# Peer review of "Is Tissue Still the Issue? The Promise of Liquid Biopsy in Uveal Melanoma"

_biomedicines, 2022, doi:10.3390/biomedicines10020506_

Round 1

Reviewer 1 Report

This review nicely summarized blood-based liquid biopsy in uveal melanoma, covering recent studies in different compositions of liquid biopsy such as CTCs, ctDNA, EVs, miRNA, proteomics, and metabolomics. Following discussion addressed medical imaging utility in staging, surveillance, and prognostication of UM, with clarifying the limitations. Most importantly, they proposed the strategy that combing liquid biopsy modalities using tumor biomarkers in the future clinical management of UM.

Obviously, the author team composes the expertise from both molecular profiling of liquid biopsy and clinicians. They focused on the critical unmet clinical challenges and proposed a multi-modality management strategy.

Minor comments are listed here.

Page 5, Line15-16: Since then, …………has progressed in UM. Should specify these two papers indicated the progress in UM, not in pan-cancer field.

Page 10, Line 17: “Different malignancies can provide varying ctDNA fragments,…” I assume this statement is regarding the fragment length? Any established reference for this statement?

Page 10, Line 23: “However, 10% tumor fraction is not feasible….”. Should add “in blood” and “for organ-confined/or intraocular UM”. And “meaning further short fragments enrichment is necessary” since enrichment could target other factors.

Author Response

Dear reviewer,

Thank you for your concise and flattering summary and we are very pleased that the goals for publishing this review came across well.

Furthermore, thank you for  the valuable suggestions for improving this manuscript.

On page 5 line 15/16 we have added:

“Since then, CTC isolation and characterization has progressed in UM.”

We regret that we have been unclear in our sentence on page 10 line 17, we have revised the sentence and added adequate references:

“Different malignancies can provide varying proportions of short (ctDNA) fragments in cfDNA[25,77]. Thus, the fragment size distribution of ctDNA in blood of UM-patients should be elucidated for enrichment prior to genotyping.”

Thank you for pointing out the vagueness of our sentence on page 10, line 23. We have revised the sentence:

“However, 10% tumor fraction in blood is not feasible for UM confined to the eye [26,70,76], meaning further enrichment of short fragments is necessary in UM.”

Thank you.

Reviewer 2 Report

A few concerns that need minor changes.

  • Some headings are lengthy especially circulating tumor cells (CTC), which can be shortened so that readers do not get bored
  • A lot of abbreviations are used, so authors should add a separate list of acronyms used in the article
  • Minor spelling mistakes
  1. Line 30 of page 4 – biopsy is a contradiction in
  2. Line 13, page-15 – lung adenocarcinoma (space is needed)
  3. Line 15, page 15 – lung cancer (space is needed)

Author Response

Dear reviewer,

We understand that some of the headings are longer and therefore we have revised the following headings:

Page 6, line 1: ‘Veridex Cellsearch’ into “Cellsearch”.

Page 7, line 1: ‘Miltenyi MACS’ into “MACS”.

Page 7, line 9: ‘ThermoFisher Dynabeads’ into “Dynabeads”.

Page 7, line 35: ‘CTC Copy number variation profiling and sequencing’ into “CTC genotyping”.

Thank you for your valuable input to add a separate list of abbreviations, which we have included on page 18 (starting from line 29).

Furthermore, we are appreciative for noticing our spelling mistakes and we have corrected them.

Thank you.